# DiverseAR: Boosting Diversity in Bitwise Autoregressive Image Generation

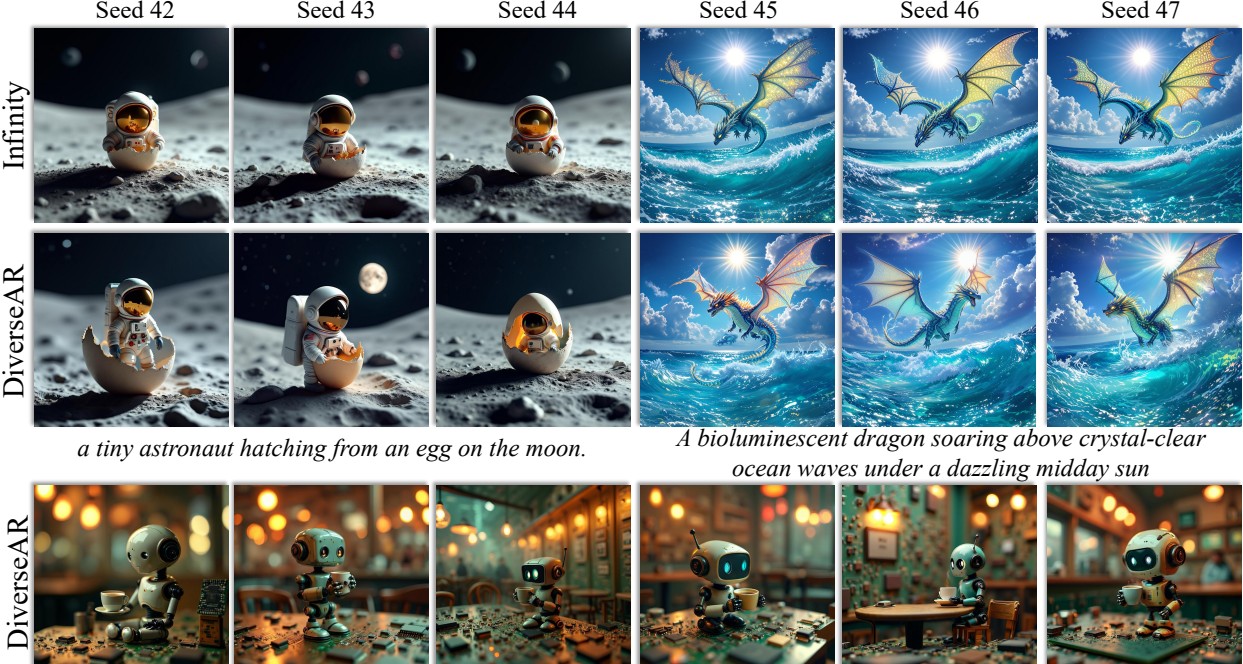

Seed 42    Seed 43    Seed 44    Seed 45    Seed 46    Seed 47

*a tiny astronaut hatching from an egg on the moon.*

*A bioluminescent dragon soaring above crystal-clear ocean waves under a dazzling midday sun*

*A tiny vintage robot sipping coffee on a circuitboard café, surrounded by microchips, resistors, and glowing transistors, moody bokeh lights, tilt-shift perspective, Wes Anderson-inspired, warm color palette.*

Figure 1: **High-resolution and diverse image synthesis results from DiverseAR**, fully unleashing the potential of bitwise autoregressive generative models.

## Abstract

In this paper, we investigate the underexplored challenge of sample diversity in autoregressive (AR) generative models with bitwise visual tokenizers. We initially analyze the factors limiting diversity in bitwise AR models and identify two key issues: **1)** the binary classification nature of bitwise modeling, which restricts the prediction space, and **2)** the overly-sharp logits distribution, which causes sampling collapse and reduces diversity. Built on these insights, we propose **DiverseAR**, a principle and effective method that enhances image diversity without sacrificing visual quality. Specifically, we introduce an adaptive logits distribution scaling mechanism that dynamically adjusts the sharpness of the binary output distribution across different sampling steps, resulting in a smoother prediction distribution and improved diversity. To mitigate the potential fidelity loss caused by distribution smoothing, we further develop an energy-based generation path search algorithm that avoids sampling low-confidence tokens, thereby preserving high visual quality. Extensive experiments highlight that DiverseAR can unlock greater diversity in bitwise autoregressive image generation.

## 1 Introduction

Recently, autoregressive (AR) models have attracted considerable attention in visual generation. Inspired by the remarkable success of large language models (Brown et al., 2020; Radford et al., 2018; Touvron et al., 2023; Achiam et al., 2023), researchers have begun to explore AR-based approaches for visual synthesis (Sun et al., 2024; Tian et al., 2024; Yu et al., 2025; Pang et al., 2024), aiming to leverage their strong modeling capacity and unified generation

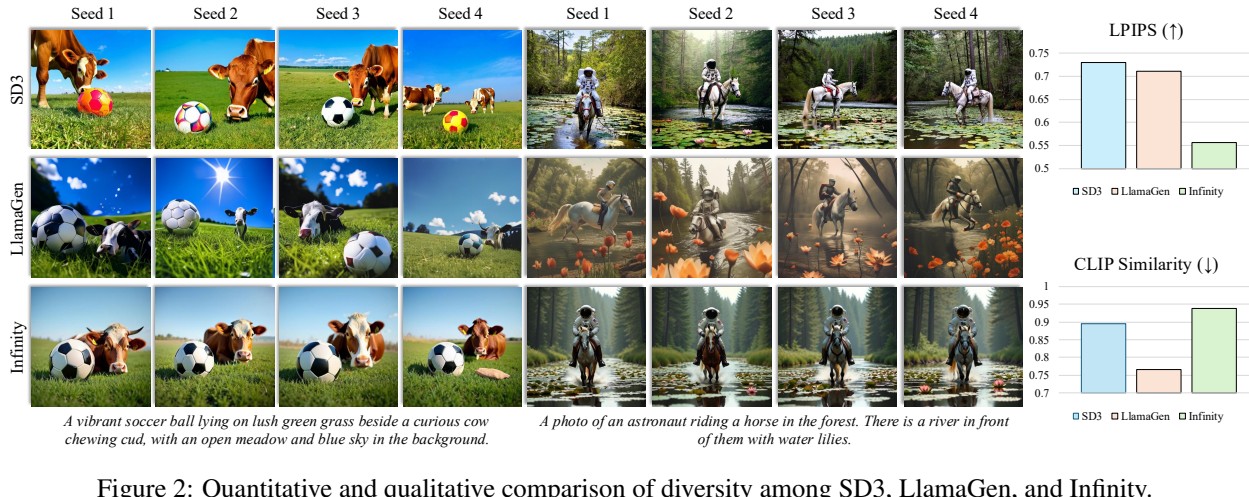

Figure 2: Quantitative and qualitative comparison of diversity among SD3, LlamaGen, and Infinity.

paradigm. Benefiting from these strengths, AR-based models have demonstrated impressive capabilities in image generation (Tian et al., 2024; Han et al., 2024; Sun et al., 2024), achieving competitive performance compared to diffusion-based (Podell et al., 2023; Chen et al., 2023) approaches in recent studies.

Existing autoregressive models for visual generation commonly adopt vector quantization (VQ) to transform continuous image representations into discrete token sequences, serving as the foundation for autoregressive modeling. Early studies (Sun et al., 2024; Pang et al., 2024; Tian et al., 2024) follow this approach by encoding images into index-based token sequences using a visual tokenizer (Razavi et al., 2019; Van Den Oord et al., 2017; Esser et al., 2021; Lee et al., 2022), and then applying AR models to generate images either token-by-token or scale-by-scale. However, this discretization process introduces quantization errors due to the limited size of the token vocabulary, hindering the generation of fine-grained details. Moreover, coarse supervision and train–inference mismatch during generation exacerbate visual degradation (Han et al., 2024), leading to artifacts and making the tokenizer a key bottleneck in AR models. To address these limitations, recent studies (Han et al., 2024) explore bitwise modeling, which replaces index-wise tokens with bitwise tokens. This design allows for an effectively unlimited token space while maintaining computational and memory efficiency. Bitwise modeling also provides finer supervision and more stable training dynamics, contributing to improved generation quality. Despite these advantages, bitwise autoregressive models exhibit limited output diversity. As illustrated in Fig. 2, Infinity (Han et al., 2024) generates significantly less diverse samples than SD3 (Esser et al., 2024) and LlamaGen (Sun et al., 2024) when sampling with different random seeds. This limitation remains under-explored, hindering the broader applicability of bitwise AR models.

In this paper, we pioneer the investigation into the diversity limitations of bitwise autoregressive models. As a first step toward a comprehensive understanding, we analyze the underlying causes of low sample diversity. Our study identifies two primary contributing factors: **1) The binary classification characteristics of bitwise modeling.** Since each bit is predicted independently as either 0 or 1, the model is inherently limited to two candidate outcomes per position. This severely constrains the sampling space, rendering top-$k$ sampling ineffective and limiting the overall expressive capacity during sampling. **2) Overconfident output distributions.** The probability distribution over the two possible bit values is often highly peaked, with one bit having significantly higher probability than the other. This causes top-$p$ sampling to frequently collapse to the most probable class, resulting in overly localized sampling and reduced exploration of alternative outcomes.

Building on these insights, we propose **DiverseAR**, an effective approach that enhances image diversity without sacrificing visual quality. As shown in Fig. 3, early coarse scales in the generation process tend to produce structurally homogeneous outputs. To address this, we introduce an **adaptive logits scaling** mechanism at coarse sampling stages, which dynamically adjusts the sharpness of the binary output distribution across sampling steps. By preventing overly confident predictions, this approach preserves uncertainty in early stages and increases the entropy of the predictive distribution. As a result, the model is encouraged to explore a broader set of plausible generation paths, leading to improved sample diversity. *However, we observe that smoothing the distribution can shift the probability mass away from the model's learned distribution, introducing local artifacts.* To mitigate this issue, we further design an **energy-based generation path search algorithm** that steers sampling away from low-probability tokens. By constraining sampling to high-confidence regions of the model's output distribution, it reduces the risk of accumulating unlikely bit patterns that can lead to artifacts, thereby preserving high visual quality.

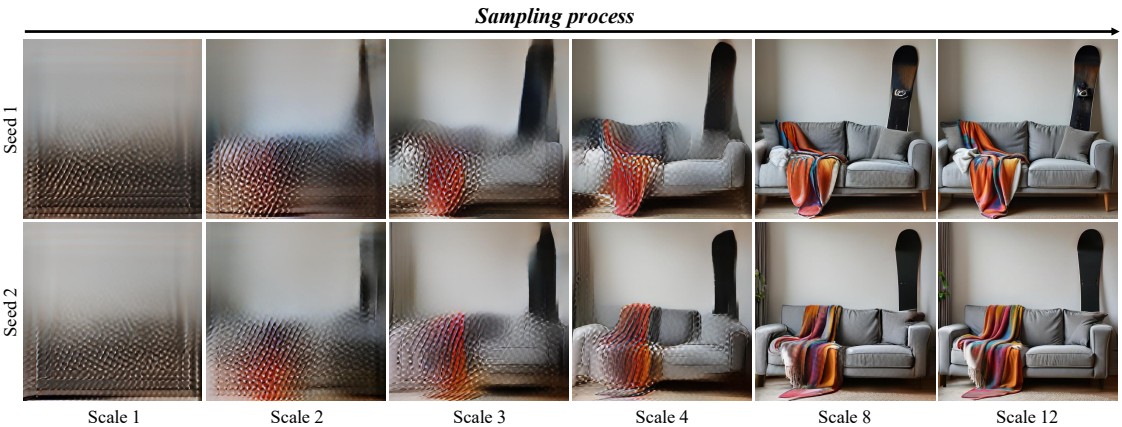

Figure 3: Visualization of sampling process for the same prompt across different random seeds.

We conduct a comprehensive experimental evaluation of our approach. The results demonstrate that DiverseAR significantly improves sample diversity while maintaining high visual fidelity, as shown in Fig. 1. Our contributions are summarized as follows:

- We present the first in-depth analysis of the diversity limitations in bitwise autoregressive models, identifying two core factors: the binary classification characteristics of bitwise modeling and the excessively peaked output distribution.

- We introduce **DiverseAR**, which combines an adaptive logits scaling mechanism with an energy-based generation path search algorithm. This design jointly enhances sample diversity while maintaining high-fidelity image synthesis.

- Extensive experiments demonstrate the superiority of our proposed method. For example, on Infinity-2B, our method improves LPIPS by 20% compared to the baseline, and achieves approximately a 5% gain in GenEval Score.

## 2 RELATED WORK

**Autoregressive Image Generation with Vector quantization.** Inspired by the success of autoregressive language models (Brown et al., 2020; Radford et al., 2018; Touvron et al., 2023; Achiam et al., 2023), autoregressive image generation (Ramesh et al., 2021; Chang et al., 2022; Yu et al., 2024; Li et al., 2024; Fan et al., 2024; Tang et al., 2024; Sun et al., 2024; Tian et al., 2024; Han et al., 2024) has advanced rapidly through the use of quantized tokenizers (Van Den Oord et al., 2017; Razavi et al., 2019; Esser et al., 2021) that embed images into compact latent spaces. Vector-quantization (VQ)-based methods (Razavi et al., 2019; Van Den Oord et al., 2017; Esser et al., 2021; Lee et al., 2022) convert image patches into discrete tokens represented by indices and employ a decoder-only transformer to predict the next-token index, resulting in efficient yet expressive image representations. Approaches like LlamaGen (Sun et al., 2024) and Parti (Yu et al., 2022) incorporate jointly learned discrete token vocabularies into transformer architectures, enabling high-quality image generation and maintaining strong scaling performance. Frameworks such as VAR (Tian et al., 2024) and FAR (Yu et al., 2025) employ coarse-to-fine sequential generation, with VAR progressively refining across spatial resolutions and FAR across frequency bands, demonstrating robust scalability. (Yu et al., 2024) propose compressing images into one-dimensional sequences, reducing redundancy and achieving more compact representations. (Guo et al., 2025) introduce a coarse-to-fine token prediction strategy, wherein the model first predicts coarse-grained indices followed by fine-grained ones.

**Autoregressive Image Generation without Vector quantization.** Finite Scalar Quantization (FSQ) (Mentzer et al., 2023) proposes quantizing tokens to constants nearest to codebook entries, which improves codebook utilization and simplifies training. Lookup Free Quantization(LFQ) (Yu et al., 2023) and Binary Spherical Quantization (BSQ) (Zhao et al., 2024) adopt binary quantization to further enhance training stability and reduce quantization error. Infinity (Han et al., 2024) employs BSQ (Zhao et al., 2024) and introduces a bitwise infinite-vocabulary classifier (IVC), enhance scalability and minimize information loss from discretization, while also integrating a bitwise self-correction mechanism to mitigate cumulative errors during autoregressive decoding. Furthermore, recent research (Tang et al., 2024; Li et al.,

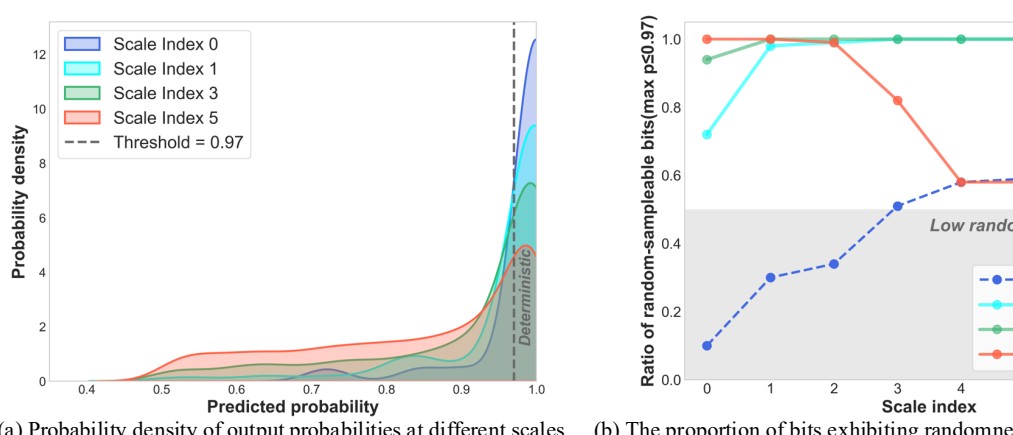

(a) Probability density of output probabilities at different scales  (b) The proportion of bits exhibiting randomness across different scales

Figure 4: Analysis of the Distribution of Predicted Logits for Binary Classifiers

2024; Ren et al., 2025; Chen et al., 2024; Fan et al., 2024) has explored combining diffusion and autoregressive models by modeling continuous tokens, substituting categorical cross-entropy with diffusion-based losses (Fan et al., 2018; Holtzman et al., 2019).

## 3 METHODOLOGY

### 3.1 PRELIMINARY

Existing autoregressive models typically adopt vector quantization (VQ) to discretize continuous images into token sequences, which are then synthesized by transformers in a causal manner, either token-by-token or scale-by-scale. Early methods often use a visual tokenizer to encode images into index-based token sequences. For instance, VAR adopts VQGAN with a multi-scale quantization layer to tokenize images and predicts residual features $\mathbf{F}_k \in [V_d]^{h_k \times w_k}$ at $k$-th scale using a $V_d$-class classifier.

However, index-wise tokenization is constrained by the limited vocabulary size, incurs quantization errors, and suffers from fuzzy supervision, causing visual detail loss and local distortions.

To address these limitations, recent work has investigated bitwise modeling, replacing index-based tokens with bitwise tokens to enhance expressiveness and reduce quantization artifacts. Infinity is one of the most notable approaches in this area. It introduces a bitwise autoregressive model, comprising a bitwise visual tokenizer, a bitwise infinite-vocabulary classifier (IVC), and a bitwise self-correction module. The IVC employs $d$ binary classifiers in parallel (where $d = \log_2(V_d)$) to predict residual features. At each scale $k$, given the token index $l$, the IVC predicts the logits $T_k^{(l,i)} : \{0, 1\} \to \mathbb{R}$ for the $i$-th bit of the $l$-th token. We then sample the bit-wise token $\mathbf{Y}_k^l$ as follows:

$$\mathbf{Y}_k^l = \left[ Y_k^{(l,1)}, Y_k^{(l,2)}, \ldots, Y_k^{(l,d)} \right], \quad Y_k^{(l,i)} \sim \text{softmax}\left( T_k^{(l,i)}/\tau \right), \quad Y_k^{(l,i)} \in \{0, 1\}. \tag{1}$$

Here, the sampling operator $\sim$ can be instantiated as $\arg\max$, top-$k$, or top-$p$ sampling. Compared to conventional classifiers, IVC is much more efficient in terms of both parameters and memory, and benefits from more steady supervision. In this work, we build upon Infinity to explore strategies for enhancing the diversity and improving the quality of bitwise autoregressive models.

### 3.2 WHY DOES BITWISE AR MODEL DEGRADE DIVERSITY?

Despite achieving impressive performance in text-to-image synthesis, the images generated by bitwise AR model exhibit limited diversity, as evidenced in Fig. 2. Through the visual analysis of the generation process, we find that at the early, coarse scales, the synthesized results already exhibit a high degree of structural homogeneity, as illustrated in Fig. 3. This observation suggests that the lack of diversity may stem from the collapse of classifier predictions in the early stages of generation.

This behavior can be traced to the design of the infinite-vocabulary classifier used in Infinity, which is composed of $d$ independent binary classifiers, as presented in Eq. 1. The binary nature of these classifiers imposes inherent constraints

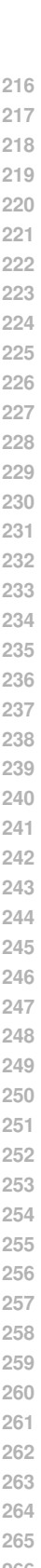

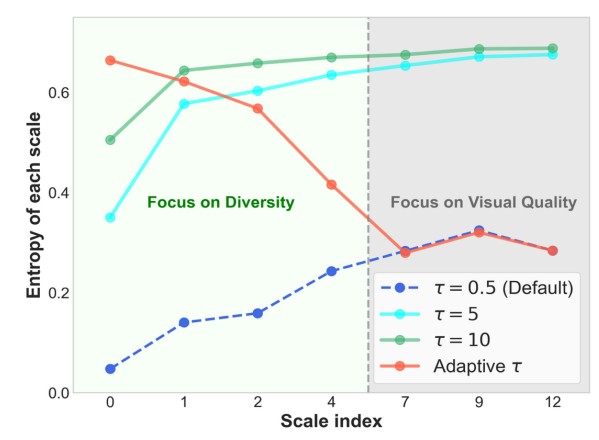

$\tau = 0.5$ (Default) — $\tau = 5$

$\tau = 10$ — Adaptive $\tau$ scaling

(a) Comparison of visual quality under different settings

(b) Entropy of the predicted probabilities at each scale under different settings

Figure 5: Quality and entropy comparison under different $\tau$ settings.

on sampling. In particular, it renders top-$k$ sampling ineffective, as each bit has only two possible outcomes. As a result, Infinity adopts top-$p$ sampling to introduce stochasticity into the generation process. To further understand the source of diversity collapse, we analyze the behavior of these binary classifiers by visualizing the distribution of their predicted logits. As shown in Fig. 4(a), the predicted probabilities are often highly peaked, with one class (either 0 or 1) receiving near-certain confidence—frequently exceeding the default top-$p$ threshold of 0.97. This overconfidence leads to a collapse in randomness: despite the use of top-$p$ sampling, the dominant class is almost always selected, effectively reducing the sampling process to a deterministic decision. Moreover, at earlier scales, this phenomenon becomes even more pronounced.

This leads to top-p sampling frequently collapsing to the class with the higher probability, thereby losing randomness. As depicted in Fig. 4 (b), under the default sampling configuration, only about $10\%$ of the bits on the first scale exhibit randomness. Moreover, this collapse of randomness at the bit level leads to constrained feature variation across sampling trajectories, ultimately resulting in reduced diversity in the generated outputs. These findings indicate that the diversity degradation in bitwise autoregressive models primarily stems from two factors: ***the binary classification nature of bitwise modeling and the overconfidence of the predicted output distributions.***

### 3.3 ADAPTIVE TEMPERATURE SCALING FOR ENHANCED DIVERSITY

Building upon these insights, a straightforward solution is to increase the temperature coefficient $\tau$ in the binary classifier (Eq. 1), which smooths the binary probability distributions and improve the effectiveness of top-$p$ sampling.

However, due to substantial variation in the predicted bit-wise logits, a fixed temperature $\tau$ may fail to provide appropriate smoothing. In cases where the logits are overly sharp, achieving desired smoothing requires a large temperature, which in turn introduces excessive randomness during the refinement of fine-grained details and ultimately degrades visual quality. As shown in Fig. 5, increasing $\tau$ to 5 or 10 results in large entropy shifts across scales compared to the default, leading to incoherent and visually distorted outputs.

To mitigate the drawbacks of simply increasing the temperature coefficient to a fixed $\tau$, we propose an adaptive temperature scaling strategy that determines $\tau_k$ for each scale $k$ based on the predicted logits, thereby achieving proper smoothing, as shown in Fig. 6(a).

Specifically, we first compute the maximum bit-probability $p_k^{(l,i)}$ for the logits $T_k^{(l,i)}$ of the $i$-th bit in the $l$-th token at scale $k$:

$$p_k^{(l,i)} = \max_{c \in \{-1,1\}} \frac{\exp\left(T_k^{(l,i)}(c)/\tau_k\right)}{\sum_{c' \in \{-1,1\}} \exp\left(T_k^{(l,i)}(c')/\tau_k\right)}. \tag{2}$$

We then compute the average of these max-probabilities across all $d$ bits for all $L_k$ tokens at scale $k$:

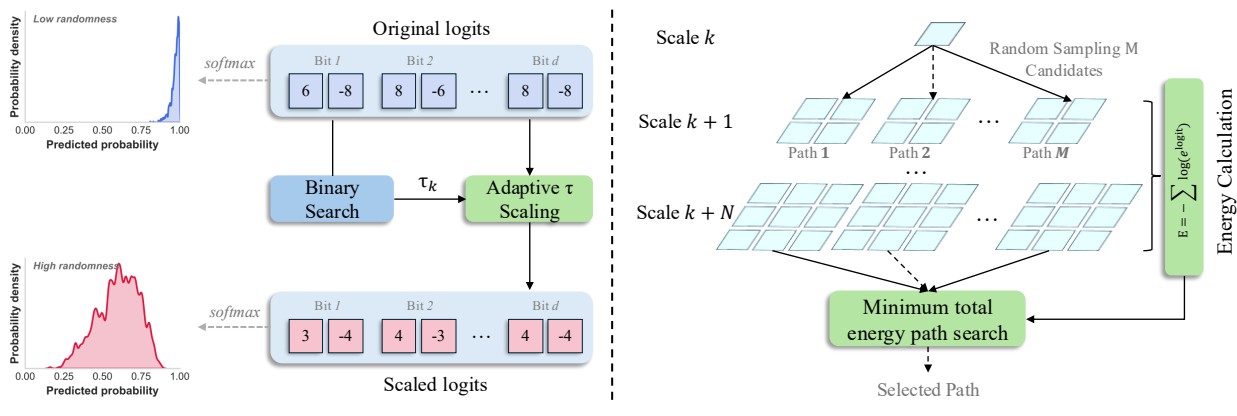

(a) Adaptive Temperature Scaling

(b) Energy-Based Generation Path Search

Figure 6: Overview of the proposed method *DiverseAR*, which consists of Adaptive $\tau$ Scaling and Energy-Based Generation Path Search.

$$\bar{p}_k(\tau_k) \;=\; \frac{1}{L_k \times d} \sum_{l=1}^{L_k} \sum_{i=1}^{d} p_k^{(l,i)}. \tag{3}$$

Intuitively, $\bar{p}_k(\tau_k)$ captures the average peak confidence of the classifier at scale $k$. To control this confidence level, we define a target smoothing level $S_k$ for each scale and search for a $\tau_k$ such that $\bar{p}_k \approx S_k$. This is efficiently achieved via binary search:

$$\left| \bar{p}_k(\tau_k) - S_k \right| < \epsilon, \tag{4}$$

where $\epsilon$ is a small numerical tolerance. The algorithmic details are provided in Appendix A. In the early diversity-oriented synthesis phase, we select smaller $S_k$ values, leading to larger $\tau_k$ values and smoother probability distributions . In the later visual refinement phase, a smaller temperature $\tau_k$ is restored to maintain the visual quality of generated images. As shown in Fig. 5(b), the adaptive temperature scaling mechanism introduces sufficient randomness in the early sampling stage (indicated by higher entropy), promoting diverse layouts, while avoiding excessive stochasticity in the later stage (lower entropy), thereby minimizing negative impacts on perceptual quality.

### 3.4 ENERGY-BASED GENERATION PATH SEARCH FOR QUALITY ENHANCEMENT

Expanding the sampling space in the early stage of image generation may lead to sampling from low-confidence regions, thereby introducing semantic artifacts into partial samples, as illustrated in Fig. 9. Prior work (Liu et al., 2020) shows that lower energy values correspond to higher logits assigned to predicted bits, indicating greater model confidence at each token position. Building on this insight, we find that in the bitwise AR model, lower energy in the logits (i.e., higher confidence) is often associated with better visual quality, as shown in Fig. 7. Motivated by this connection, we propose an energy-based generation path search algorithm, as illustrated in Fig. 6(b).

Specifically, we follow the definition of energy proposed in (Liu et al., 2020). At the $k$-th scale, the energy of its predicted logits can be computed using the following formulation:

$$E_k = -\frac{1}{L_k} \sum_{l=1}^{L_k} \log \left( \sum_{i=1}^{d} e^{T_k^{l,i}} \right), \tag{5}$$

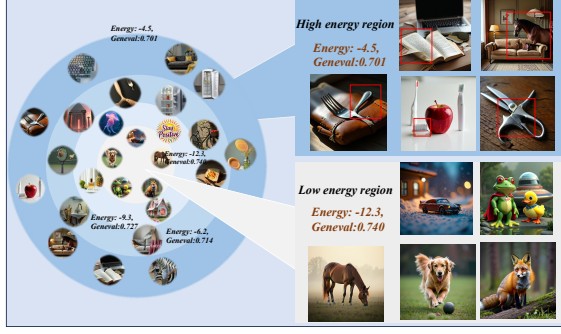

Figure 7: Visualization results of different energy regions

where $T_k^{l,i}$ denotes the logit for the $i$-th bit of the $l$-th token at scale $k$, and $L_k$ is the total number of tokens at that scale. After the adaptive temperature scaling at scale $k$, we perform a generation path search to identify low-energy trajectories. We first sample $M$ candidate initialization and propagate each forward through the next $N$ scales, yielding $M$ distinct candidate paths. For the $m$-th path, we compute its cumulative energy by averaging per-scale energy values:

$$E^m = \frac{1}{N} \sum_{j=k+1}^{k+N} E_j^m, \quad m = 1, 2, \ldots, M. \tag{6}$$

We then select the path with the lowest cumulative energy by solving:

$$r^* = \underset{m \in \{1, \ldots, M\}}{\arg\min} \ E^m. \tag{7}$$

The selected path $r^*$ is then propagated through the remaining scales to complete the sampling process. Notably, since this selection is performed in early stages, where both resolution and token count are relatively low, the additional computational overhead is minimal.

## 4 EXPERIMENTS

Table 1: Diversity and Quality Evaluation: LPIPS and CLIP Similarity, GenEval and DPG Benchmark

| Model | Diversity | | GenEval(↑) (Ghosh et al., 2023) | | | | DPG(↑) (Hu et al., 2024) | | |
|---|---|---|---|---|---|---|---|---|---|
| | LPIPS(↑) | CLIP(↓) | Two Obj | Position | Color Attri | Overall | Global | Relation | Overall |
| Diffusion Model | | | | | | | | | |
| SDv1.5 (Rombach et al., 2022) | 0.7909 | 0.8291 | 0.38 | 0.04 | 0.06 | 0.37 | 74.63 | 73.49 | 63.18 |
| PixArt-alpha (Chen et al., 2023) | 0.6896 | 0.9096 | 0.50 | 0.08 | 0.07 | 0.48 | 74.97 | 82.57 | 71.11 |
| SDXL (Podell et al., 2023) | 0.7403 | 0.8768 | 0.74 | 0.15 | 0.23 | 0.55 | 83.27 | 86.76 | 74.65 |
| SD3.5 -medium (Esser et al., 2024) | 0.7294 | 0.8952 | 0.74 | 0.34 | 0.36 | 0.62 | - | - | - |
| AutoRegressive Models | | | | | | | | | |
| LlamaGen (Sun et al., 2024) | 0.7110 | 0.7662 | 0.34 | 0.07 | 0.04 | 0.32 | - | - | 65.16 |
| Hart (Tang et al., 2024) | 0.7106 | 0.8834 | - | - | - | 0.52 | - | - | 80.89 |
| Show-o (Xie et al., 2024a) | 0.6427 | 0.9251 | 0.80 | 0.31 | 0.50 | 0.68 | - | - | 67.48 |
| Infinity-2B (Han et al., 2024) | 0.5555 | 0.9381 | 0.83 | 0.44 | 0.53 | 0.716±0.05 | 88.61 | **87.97** | 81.51±0.3 |
| DiverseAR-2B | 0.6712 | 0.9192 | **0.88** | **0.51** | **0.60** | **0.760±0.04** | 89.20 | 87.57 | **81.72±0.3** |
| Infinity-8B (Han et al., 2024) | 0.3745 | 0.9583 | 0.90 | 0.62 | **0.69** | 0.797±0.02 | 86.93 | **91.24** | 85.88±0.2 |
| DiverseAR-8B | 0.5510 | 0.9354 | **0.91** | **0.63** | 0.68 | **0.802±0.03** | 92.79 | 90.55 | **86.14±0.2** |

### 4.1 EXPERIMENTAL SETTINGS

**Evaluation Metrics.** To evaluate diversity, we use 50 prompts and generate 50 images per prompt with different random seeds, resulting in a total of 2,500 images. For each prompt, we compute pairwise LPIPS (Zhang et al., 2018) and CLIP (Radford et al., 2021) similarities among the 50 samples, average these values over all pairs to obtain a per-prompt score, and then report the mean across prompts as the final diversity scores. For quality assessment, we report GenEval (Ghosh et al., 2023), DPG (Hu et al., 2024), ImageReward (Xu et al., 2023), and HPSv2 (Wu et al., 2023) scores. GenEval and DPG scores are computed across multiple seeds to estimate error bars. The results of ImageReward and HPSv2 are reported in the appendix C.

**Implementation Details.** For Infinity-2B, we use the default setting with a CFG of 4 and a fixed sampling temperature of 0.5. DiverserAR-2B sets CFG to 4 and applies an adaptive temperature schedule to target average maximum bit probabilities $S_k$ that increases linearly from 0.60 to 0.90 across the first half of the scales. For the remaining ones, we use argmax sampling to select the highest-probability bit at each position. During energy-based path search, we sample $M = 8$ candidate paths at scale 2, propagate each through scales 3 to 6, and compute the average cumulative energy along each path. The details for the 8B model configuration are provided in Appendix B.1. To verify the scalability of our method, we also evaluate it on the VQ-based autoregressive model HART. The experimental details are provided in Appendix C.1. All experiments are run on NVIDIA H20 GPUs.

### 4.2 MAIN RESULTS

**Quantitative Results.** Tab. 1 compares the diversity and quality metrics across different methods. Compared to Infinity-2B (Han et al., 2024), DiverseAR-2B improves LPIPS by 0.1216 (approximately 20%) and decreases CLIP

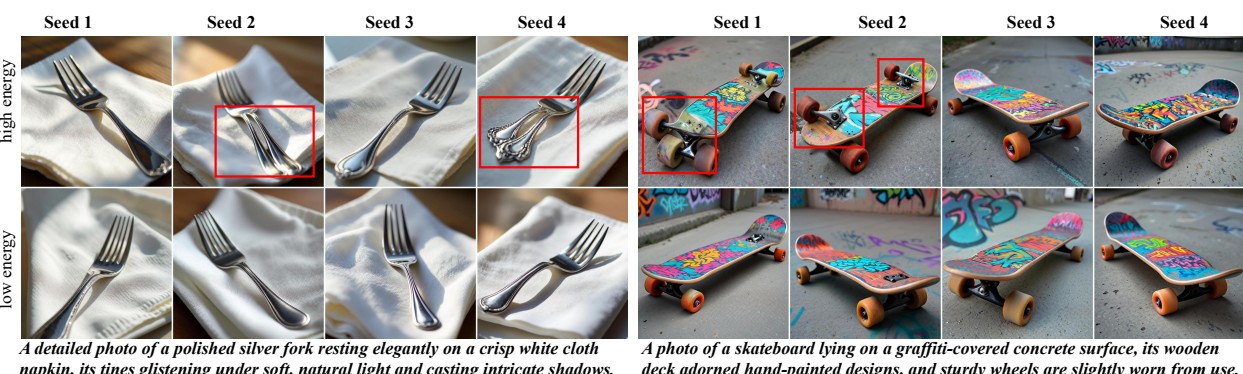

Figure 8: Comparison of output diversity between the original method and our approach

Figure 9: Quality Comparison of High-Energy vs. Low-Energy Sampling Outputs

similarity by 0.0213, yielding the diversity level comparable to that of SD3.5 and LlamaGen. In terms of quality, our method consistently improves GenEval and DPG scores. The "Position" and "Color Attribution" sub-scores in GenEval each increase by 7%, and the overall score improves by 4.4%. We also report the comparison results for the 8B model in Appendix E. Our method achieves substantial gains—LPIPS improves by approximately 60%—while still preserving high visual quality. Furthermore, we also evaluate our method on an additional VQ-based autoregressive model, HART (Tang et al., 2024). The corresponding results are provided in Appendix C.1.

Table 2: Diversity and Quality Evaluation under Different Search Strategies in the 2B model

| Method | Latency | Diversity | | GenEval | DPG |
|---|---|---|---|---|---|
| | | LPIPS | CLIP | Overall | Overall |
| Baseline | ×1.0000 | 0.5555 | 0.9381 | 0.716 | 81.51 |
| + Adaptive $\tau$ | ×1.0005 | 0.6768 | 0.9172 | 0.739 | 81.56 |
| + Energy search | ×1.1187 | 0.5426 | 0.9398 | 0.744 | 81.58 |
| + Adaptive $\tau$ & Energy search | ×1.1192 | 0.6712 | 0.9192 | **0.760** | **81.72** |

Table 3: Comparison of Different $\tau$ Settings for Diversity and Quality Evaluation in the 2B model

| Metric | $\tau = 5$ (half) | $\tau = 10$ (half) | $\tau = 20$ (half) | **Adaptive $\tau$** |
|---|---|---|---|---|
| LPIPS | 0.6767 | 0.7132 | **0.7578** | 0.6712 |
| CLIP | 0.9176 | 0.8874 | **0.8130** | 0.9192 |
| GenEval | 0.728 | 0.704 | 0.593 | **0.760** |

**Qualitative Results.** Fig. 8 visualizes output diversity across different random seeds, comparing our method with the baseline in 2B models. Additional comparisons are provided in Appendix E. These results demonstrate the effectiveness and superiority of our proposed method, which achieves significantly enhanced diversity while maintaining high visual quality.

### 4.3 ABLATION STUDIES

**The impact of individual components.** Tab. 2 presents the impact of each component in our method on final performance. As shown, adaptive temperature scaling significantly improves the diversity of generated images (LPIPS: $0.5555 \rightarrow 0.6712$) while maintaining high visual quality and introduces only negligible extra inference time. Meanwhile, by combining the energy-based generation path search, we achieve the best visual quality (GenEval: $0.716 \rightarrow 0.760$).

Moreover, since the path search operates only at the coarser early scales, it introduces only minimal latency($\times 1 \to \times 1.1192$).

Table 4: Sensitivity analysis of $S_k$: effect of different schedules on diversity and quality.

| Method | LPIPS ↑ | CLIP ↓ | GenEval ↑ |
|---|---|---|---|
| Baseline | 0.5555 | 0.9381 | 0.716 |
| Fixed $S_k$ ($S_k = 0.6$) | **0.6913** | **0.9021** | 0.750 |
| Fixed $S_k$ ($S_k = 0.65$) | 0.6801 | 0.9137 | 0.751 |
| Fixed $S_k$ ($S_k = 0.7$) | 0.6653 | 0.9226 | 0.751 |
| Linear $S_k$ ($0.6 \to 0.9$) | 0.6712 | 0.9192 | **0.760** |

Table 5: Effect of varying the number of selected scales on diversity and GenEval scores under linear adjustment.

| Metric | Number of Selected Scales | | | | |
|---|---|---|---|---|---|
| | 0 | 1 | 3 | 5 | 7 |
| LPIPS | 0.5555 | 0.5954 | 0.6669 | 0.6713 | **0.6712** |
| CLIP | 0.9381 | 0.9283 | 0.9188 | 0.9182 | **0.9192** |
| GenEval (Adaptive $\tau$) | 0.716 | 0.719 | 0.725 | 0.733 | **0.739** |
| GenEval (Adap. + Search) | 0.744 | 0.748 | 0.752 | 0.755 | **0.760** |

**Comparison of Fixed and Adaptive $\tau$.** Tab. 3 compares three fixed $\tau$ settings ($\tau = 5, 10, 20$ applied to the first half of scales) with our adaptive $\tau$. We observe that simply increasing the early-stage $\tau$ does not effectively balance diversity and quality: although $\tau = 20$ yields a significant gain in diversity, it also causes a notable drop in quality. The visualization of different $\tau$ settings is shown in Fig. 10. Furthermore, due to the substantial variation in logits distributions across samples, selecting a single fixed $\tau$ that works universally is difficult. By contrast, determining $\tau$ adaptively through the target average peak confidence $S_k$ provides a more robust and effective solution.

**Sensitivity Analysis of $S_k$.** Tab. 4 presents the sensitivity analysis of $S_k$ on the final performance of our method. The results show that DiverseAR consistently outperforms the baseline across a broad range of parameter settings without requiring extensive or fine-grained tuning. For example, using a fixed $S_k = 0.6$ achieves stronger diversity with an LPIPS score of 0.6913, compared to the baseline of 0.5555. At the same time, the linear $S_k$ schedule yields a GenEval score of 0.760, which demonstrates a more stable and effective balance between diversity and quality. These observations indicate that the improvements mainly stem from the core design of our method—namely, adjusting the overly sharp probability distributions in the early stages to enhance diversity.

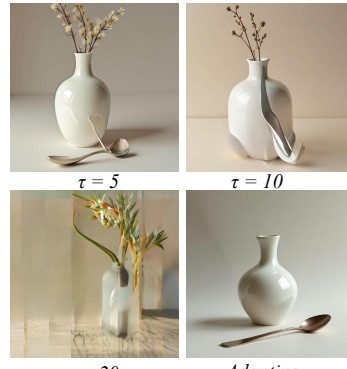

*a photo of a vase and a spoon*

Figure 10: Visualization results of different $\tau$ settings

**The impact of varying the number of adaptive scaling layers.** Tab. 5 presents LPIPS and GenEval results when adaptive temperature scaling is applied to an increasing number of initial scales. Even without any distribution adjustment (i.e., using only energy-based search), GenEval improves by 0.028. As more scales undergo temperature adjustment, GenEval rises progressively, and combining adaptive scaling with energy-based search yields additional gains. Since the target smoothing level reaches $S_7 = 0.9$ at the 7th scale, we apply argmax sampling for the remaining scales without further adjusting the logit distribution.

We also provide additional ablation studies, including the computational overhead of energy-based search on different models, the GenEval scores under different search metrics, the diversity comparison across different CFG scales, as well as other related analyses. More experimental results and ablations are reported in Appendix C.

## 5 CONCLUSION

In this work, we conduct the first in-depth investigation into the diversity limitations of bitwise autoregressive models for image generation. Through detailed analyses, we identify two key factors that restrict sample diversity: the binary nature of bitwise modeling, which narrows the sampling space, and the overly peaked output distribution, which causes sampling collapse and suppresses variability. To address these challenges, we propose ***DiverseAR***, a simple yet effective method that enhances diversity without compromising visual fidelity. Our approach introduces an adaptive logits scheduling mechanism to maintain uncertainty across early sampling stages and an energy-based generation path search algorithm to avoid low-confidence predictions. Extensive experiments on multiple benchmarks validate the effectiveness of DiverseAR in producing more diverse and high-quality samples, demonstrating its potential to improve the applicability of bitwise autoregressive generation.

## 6 ETHICS STATEMENT

Our work focuses on improving the diversity of bitwise autoregressive image generation models. We do not foresee direct ethical concerns beyond those commonly associated with generative models, such as potential misuse for generating misleading or harmful content. All experiments are conducted on publicly available datasets under their corresponding licenses. No personally identifiable information or sensitive data was used. We encourage responsible and transparent use of our method in downstream applications.

## 7 REPRODUCIBILITY STATEMENT

We are committed to ensuring the reproducibility of our results. All implementation details, including model configurations, and evaluation settings, are provided in Sec. 4.1 and Appendix B. We have released pseudocode for the proposed algorithms, and the setup for evaluation benchmarks. To facilitate replication, the source code and scripts for running experiments will be made publicly available upon publication.

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

APPENDIX

## A    ALGORITHMIC IMPLEMENTATION

We present the implementation details of our two core components in Algorithms 1 and 2.

Algorithm 1 outlines the Adaptive Temperature Scaling process, which dynamically adjusts the temperature $\tau_k$ at each scale to match the target average maximum bit probability $S_k$. To achieve this, we adopt a binary search strategy bounded by pre-defined minimum and maximum temperatures ($\tau_{\min} = 0.001$, $\tau_{\max} = 100$), iteratively refining $\tau_k$ until the target smoothness criterion is satisfied within a tolerance of $\epsilon = 0.005$.

Algorithm 2 presents the Energy-Based Path Search, which aims to select the most coherent generation trajectory across scales. At a designated sampling scale $k$, we first generate $M$ candidate paths under the current $\tau_k$. Each path is then propagated through the subsequent $N$ scales, and its average energy is computed. The path with the lowest cumulative energy is selected as the final decoding trajectory.

Together, these two algorithms enhance the diversity and quality of bitwise autoregressive generation.

---

**Algorithm 1** Adaptive Temperature Scaling

1: **Input:** targets $S_k$, predicted logits $T_k$, tolerance $\epsilon$, bounds $\tau_{\min}, \tau_{\max}$
2: Initialize $\ell \leftarrow \tau_{\min}, \quad u \leftarrow \tau_{\max}$
3: **repeat**
4:     $\tau \leftarrow (\ell + u)/2$
5:     Compute $\bar{p}_k(\tau)$
6:     **if** $\bar{p}_k(\tau) > S_k$ **then**
7:         $u \leftarrow \tau$
8:     **else**
9:         $\ell \leftarrow \tau$
10:    **end if**
11: **until** $|\bar{p}_k(\tau) - S_k| < \epsilon$
12: $\tau_k \leftarrow \tau$
13: **Output:** predicted probability $\mathrm{softmax}(T_k/\tau_k)$

---

**Algorithm 2** Energy-Based Path Search

1: **Input:** sampling scale $k$, lookahead $N$ scales, candidates path number $M$
2: Generate logits at scale $k$ under $\tau_k$
3: Sample $M$ candidate generation paths
4: **for** $m = 1, \ldots, M$ **do**
5:     $E^m \leftarrow 0$
6:     **for** $j = k+1, \ldots, k+N$ **do**
7:         $E^m += E_j^m$
8:     **end for**
9:     $E^m \leftarrow E^m/N$
10: **end for**
11: $r^* \leftarrow \underset{m \in \{1, \ldots, M\}}{\arg\min} E^m$
12: **Output:** $r^*$-th generation path

---

## B    MORE EXPERIMENTAL DETAILS

### B.1    IMPLEMENTATION DETAILS OF THE DIVERSITY METRICS

We use the official LPIPS implementation with an AlexNet (Krizhevsky et al., 2017) backbone, normalizing images to $[-1, 1]$ and computing average pairwise distances across upper-triangular entries. To reduce memory usage, the computation is split into smaller chunks. For CLIP similarity, we use the Hugging Face `openai/clip-vit-base-patch32` model. Images are encoded into L2-normalized vectors via the CLIP (Radford et al., 2021; Dosovitskiy et al., 2020) image encoder, and average cosine similarities are computed from the upper-triangular portion of the similarity matrix.

### B.2    THE IMPLEMENTATION DETAILS OF DIVERSEAR-8B

For the Infinity-8B model, the baseline follows the default configuration: a CFG of 4 and a fixed sampling temperature of 1. In our DiverseAR-8B, we also use a CFG of 4 but employ an adaptive temperature schedule that drives the average maximum bit probability to $\{0.60, 0.60, 0.65, 0.65, 0.65, 0.7, 0.7, 0.7\}$ over the first eight scales. For the remaining scales, we revert to argmax sampling, selecting the highest-probability bit at each position. In our energy-based path search, at scale 3 we sample $M = 8$ candidate token sets and propagate each through scales 3–7, computing the cumulative average energy along each trajectory. We then select the lowest-energy path to complete the sample. All experiments were conducted on NVIDIA H20 GPUs.

# C ADDITIONAL EXPERIMENTAL RESULTS AND ABLATIONS

## C.1 ADDITIONAL EXPERIMENTAL RESULTS

**Human Preference Evaluation.** We further assess our method through both quantitative benchmarks and a user study. Tab. 6 reports ImageReward and HPSv2.1 scores for the 2B models, where DiverseAR outperforms the Infinity baseline, confirming improved diversity without sacrificing visual fidelity. A user preference study is also carried out following the setup of Infinity. Specifically, we developed a web interface that displays paired image grids generated by Infinity and DiverseAR side by side. Volunteers were asked to choose the better set in terms of *overall quality*, *prompt following*, and *diversity*. We presented 200 such pairs and collected evaluations from 50 participants. The entire study was conducted double-blind: participants neither knew which model produced which image nor saw others' choices during evaluation. As reported in Table 7, a majority of participants (66%) preferred DiverseAR for image quality, and nearly all (91%) rated its outputs as more diverse.

**Results on the VQ-based HART Model.** We present the results for the VQ-based autoregressive model HART. For this model, we leverage an adaptive noise injection strategy to enhance sample diversity in the early stages of sampling. Specifically, we introduce a linearly decayed Gumbel noise strength, ranging from $[1.4, 1.3, 1.2, 1.1, 1.0, 0.9, 0.8]$ in the first few sampling steps, to perturb the distribution of the logits, while keeping the remaining steps or scales consistent with the default setting. The results are reported in Tab. 8. We observe that by adjusting the probability distribution through this noise scheduling, the HART model achieves improved diversity without compromising perceptual quality.

Table 6: Human preference metrics on Infinity-2B and DiverseAR-2B.

| Method | ImageReward ↑ | HPSv2.1 ↑ |
|---|---|---|
| Infinity | 30.26 | 0.8972 |
| DiverseAR | **30.41** | **0.9013** |

Table 7: User study results: percentage of participants preferring each method.

| Method | Overall ↑ | Prompt Following ↑ | Diversity ↑ |
|---|---|---|---|
| Infinity | 0.34 | 0.47 | 0.09 |
| DiverseAR | **0.66** | **0.53** | **0.91** |

Table 8: The comparison of HART and DiverseAR on diversity (LPIPS, CLIP) and GenEval benchmarks.

| | Diversity | | GenEval | | | | |
|---|---|---|---|---|---|---|---|
| **Model** | LPIPS↑ | CLIP↓ | Two Obj | Position | Color Attri | single_object | overall |
| HART | 0.7107 | 0.8834 | 0.63 | 0.11 | 0.19 | 0.97 | 0.511 |
| HART+DiverseAR | **0.7501** | **0.8685** | 0.61 | **0.17** | **0.23** | 0.97 | **0.518** |

## C.2 ADDITIONAL ABLATIONS

Table 9: Diversity comparison between Infinity-2B and DiverseAR-2B at different CFG scales.

| CFG scale | Model | LPIPS ↑ | CLIP ↓ |
|---|---|---|---|
| 2 | Infinity-2B | 0.5581 | 0.9301 |
| 2 | DiverseAR-2B | **0.6720** | **0.9101** |
| 3 | Infinity-2B | 0.5548 | 0.9366 |
| 3 | DiverseAR-2B | **0.6738** | **0.9144** |
| 4 | Infinity-2B | 0.5555 | 0.9381 |
| 4 | DiverseAR-2B | **0.6712** | **0.9192** |

Table 10: Comparison of GenEval scores obtained under different search metrics.

| Method | GenEval ↑ |
|---|---|
| Baseline | 0.716 |
| + Negative log-probability | 0.748 |
| + Entropy search | 0.748 |
| + Energy-based search | **0.760** |

**The impact of different search metrics on GenEval Score** We investigate how various metrics correlate with GenEval performance under the default configuration. Specifically, for each metric (e.g., entropy, cumulative energy, and mean maximum bit probability), we generated 50 samples per GenEval prompt and grouped them into five percentile bins based on their respective metric scores. For each bin, we then computed the average GenEval score. As shown in Fig. 11, all three metrics exhibit a linear relationship with quality, with energy showing the strongest correlation—lower energy consistently corresponds to higher GenEval scores, indicating better image fidelity.

Tab. 10 presents several alternative search metrics to further investigate their impact on performance. Among them, energy-based search achieves the highest GenEval scores, highlighting the superiority of energy as a search criterion. This advantage arises because computing energy directly from logits provides a more faithful measure of model confidence, whereas applying softmax may discard valuable information contained in the model's raw outputs. Theoretical justification for this property can be found in (Liu et al., 2020).

Fig. 9 contrasts samples generated via low- versus high-energy decoding paths: high-energy trajectories often produce local artifacts and coherence breaks, whereas low-energy trajectories yield more coherent outputs. Accordingly, energy is employed as the criterion for guiding the generation path search.

**Impact of CFG on Diversity and Robustness of DiverseAR**    Tab. 9 reports the diversity scores (LPIPS and CLIP similarity) of Infinity-2B and DiverseAR-2B under varying CFG settings. While lowering the CFG scale typically enhances the diversity of diffusion models, it has limited effect on the bitwise autoregressive model Infinity. As shown, reducing the CFG from 4 to 2 leads to only a marginal increase in LPIPS, from 0.5555 to 0.5881. In contrast, DiverseAR consistently achieves substantial improvements in generation diversity across all CFG settings, suggesting that our method is robust and effective under different guidance strengths.

**Candidate Number $M$ and Lookahead Depth $N$**    Tab. 11 reports the effect of varying the number of candidate paths $M$ and lookahead depth $N$ on GenEval scores and time cost for both the 2B and 8B models. As shown, the performance of our method remains relatively stable across a broad range of values, indicating that DiverseAR is not highly sensitive to these hyperparameters. This robustness highlights the practicality of our default configuration, which achieves a favorable balance between computational efficiency and output quality.

**Improve diversity by more detailed prompts.**    Tab. 12 compares diversity obtained by leveraging LLMs to rewrite 30 prompts into 50 variations each (by permuting location, pose, size, and color of objects) with our proposed method. We then evaluated both approaches on a dataset of 30 samples. While prompt rewriting improves diversity to some extent (LPIPS: 0.6549), our method achieves a significantly higher score (LPIPS: 0.7133), demonstrating its clear advantage.

**Combine multiple bits forming an int token and use top-k sampling.**    We further evaluate the effect of applying top-k sampling after combining multiple bits into integer tokens. As shown in Tab. 13, this strategy yields only marginal improvements in diversity compared to the baseline, while our method significantly outperforms it. Moreover, top-k sampling incurs a substantial increase in inference time (Top-5: 1.45× vs. Ours: 1.0005×), making it less practical in comparison.

We find that the probabilities for most of the 16-bit combinations concentrate on the first five. We further visualize the predicted probabilities of the first five combinations, as shown in the Tab. 14, and observe that the distributions are extremely sharp. This limits the effectiveness of top-k sampling in increasing diversity and significantly increases inference time.

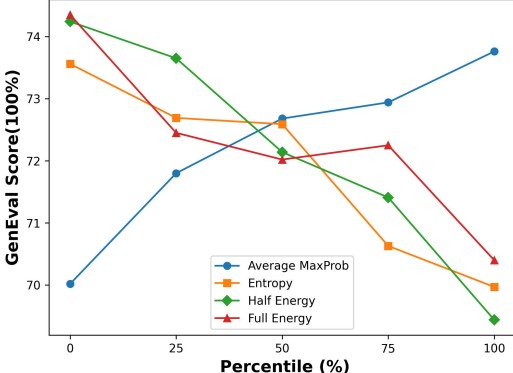

Figure 11: Relationships Among Metrics and GenEval Scores

| Setting | DiverseAR-2B | | DiverseAR-8B | |
|---|---|---|---|---|
| | GenEval ↑ | Time | GenEval ↑ | Time |
| Baseline | 0.716 | 1x | 0.797 | 1x |
| M=8,N=2 | 0.748 | 1.07x | 0.791 | 1.08x |
| M=8,N=3 | 0.751 | 1.09x | 0.797 | 1.10x |
| M=8,N=4 (default) | 0.760 | 1.12x | 0.802 | 1.13x |
| M=4,N=4 | 0.747 | 1.06x | 0.793 | 1.07x |
| M=6,N=4 | 0.754 | 1.09x | 0.798 | 1.10x |
| M=8,N=4 (default) | 0.760 | 1.12x | 0.802 | 1.13x |

Table 11: Comparison of different $(M, N)$ settings for DiverseAR-2B and DiverseAR-8B in terms of GenEval and wall-clock time.

Table 12: Comparison of prompt rewriting versus our method.

| Method | LPIPS ↑ | CLIP ↓ |
|---|---|---|
| Baseline | 0.5536 | 0.9444 |
| Rewrite | 0.6549 | 0.9166 |
| Ours | **0.7133** | **0.9065** |

Table 13: Comparison of top-k sampling (bit-to-int tokens) versus our method.

| Method | T ↓ | LPIPS ↑ | CLIP ↓ | GenEval ↑ |
|---|---|---|---|---|
| Baseline | ×1.0000 | 0.5536 | 0.9444 | 0.716 |
| Top-5 Sampling | ×1.4517 | 0.5783 | 0.9388 | 0.717 |
| Ours | ×1.0005 | **0.7133** | **0.9065** | **0.740** |

Table 14: Distribution of top-k probabilities across different scales.

| Scale | Top-1 Prob | Top-2 Prob | Top-3 Prob | Top-4 Prob | Top-5 Prob |
|---|---|---|---|---|---|
| Scale 0 | 0.62 | 0.18 | 0.11 | 0.07 | 0.003 |
| Scale 1 | 0.58 | 0.23 | 0.06 | 0.03 | 0.003 |
| Scale 2 | 0.37 | 0.18 | 0.10 | 0.05 | 0.02 |
| Scale 3 | 0.29 | 0.15 | 0.08 | 0.05 | 0.02 |

## D   DETAILS ON ADDITIONAL METRICS FOR PATH SELECTION

To further analyze the correlation between intermediate representations and final image quality, we report two additional search-related metrics in Fig. 11: entropy and average maximum bit probability.

First, we compute the unadjusted bit probability $p_k'^{(l,i)}$, i.e. the predicted probability of the $i$-th bit for token $l$ at scale $k$ before any adaptive adjustment:

$$p_k'^{(l,i)} = \max_{c \in \{-1,1\}} \frac{\exp\big(T_k^{(l,i)}(c)/\tau\big)}{\sum_{c' \in \{-1,1\}} \exp\big(T_k^{(l,i)}(c')/\tau\big)}. \tag{8}$$

Based on Eq. equation 8, the entropy at scale $k$ (before applying any temperature adjustment) is defined as:

$$H_k = \frac{1}{L_k d} \sum_{l=1}^{L_k} \sum_{i=1}^{d} \Big[ p_k'^{(l,i)} \log_2 p_k'^{(l,i)} + (1 - p_k'^{(l,i)}) \log_2(1 - p_k'^{(l,i)}) \Big]. \tag{9}$$

Similarly, the mean maximum bit probability at scale $k$ is computed as:

$$P_k = \frac{1}{L_k d} \sum_{l=1}^{L_k} \sum_{i=1}^{d} p_k'^{(l,i)}, \tag{10}$$

which reflects the average confidence across all predicted bits at the given scale.

To evaluate the $m$-th generation trajectory across scales, we define aggregated forms of these metrics over $N$ successive scales as:

$$H^m = \frac{1}{N} \sum_{j=k+1}^{k+N} H_j^m, \tag{11}$$

$$P^m = \frac{1}{N} \sum_{j=k+1}^{k+N} P_j^m. \tag{12}$$

These aggregated entropy and confidence measures can be computed for each sampled trajectory during generation. Empirically, we observe that samples with lower entropy or higher average maximum bit probability are more likely to yield higher-quality images. As shown in Fig 11, both metrics exhibit strong linear correlations with the final GenEval scores, suggesting that they can serve as reliable indicators for selecting high-fidelity outputs in the path search procedure.

## E    MORE RESULTS

We present additional qualitative results for the Infinity-2B model in Fig. 12 and 13, and for the Infinity-8B model in Fig. 14 and 15. In Fig. 12, for the first prompt, our method not only alters the layout of the samples but also their visual style. Fig. 13 offers further comparisons under 2B across additional prompts. Similarly, Fig. 14 and 15 show that under the 8B model, DiverseAR consistently yields substantially more diverse outputs. These examples demonstrate that DiverseAR enhances output variability while preserving high visual fidelity.

## F    ADDITIONAL RELATED WORK

**Diversity Control in AR Models.**    Autoregressive generation adopts decoding heuristics from language modeling to modulate diversity. Top-$k$ sampling restricts token selection to the $k$ most probable options at each step (Fan et al., 2018; Radford et al., 2019; Keskar et al., 2019; Ramesh et al., 2021), while nucleus (top-$p$) sampling selects the smallest token set whose cumulative probability exceeds $p$(Holtzman et al., 2019; Yu et al., 2022; Zhang et al., 2020). Typical sampling filters out tokens with information content deviating from the context's average uncertainty, retaining those within a tolerance range to balance quality and diversity(Meister et al., 2022). Truncation sampling treats the output as a mixture of ideal and smoothed distributions, pruning tokens below an entropy-conditioned threshold (Hewitt et al., 2022; Wang et al., 2023; Hao et al., 2024). Diverse beam search adds inter-beam penalties to mitigate mode collapse (Vijayakumar et al., 2016; Wu et al., 2022; Zhang et al., 2023), while minimum Bayes–risk (MBR) decoding selects outputs minimizing expected task-specific loss (Bertsch et al., 2023; Chang et al., 2022). There is also work (Xie et al., 2024b; Zhu et al., 2024) that adjusts the temperature coefficient to calibrate sampling in large language models. However, our approach is fundamentally different: we focus on smoothing the overly sharp early-stage distributions in bitwise autoregressive models, rather than calibrating LLM sampling.

## G    LIMITATION AND FUTURE WORK

One limitation of our method is the slight increase in inference time caused by the search over multiple candidate paths. In practice, the overhead remains limited, amounting to approximately 1.12× that of the baseline under the default setting. In future work, we will explore training-time strategies to better address the diversity limitations of the bitwise autoregressive generation model.

## H    THE USE OF LARGE LANGUAGE MODELS

In this work, we employed large language models (LLMs) to assist with minor vocabulary refinement and formatting adjustments. Specifically, we used LLMs to improve sentence logic, condense paragraph length, and adjust the formatting of tables and figures. We take full responsibility for any consequences arising from the use of LLMs in the preparation of this manuscript.

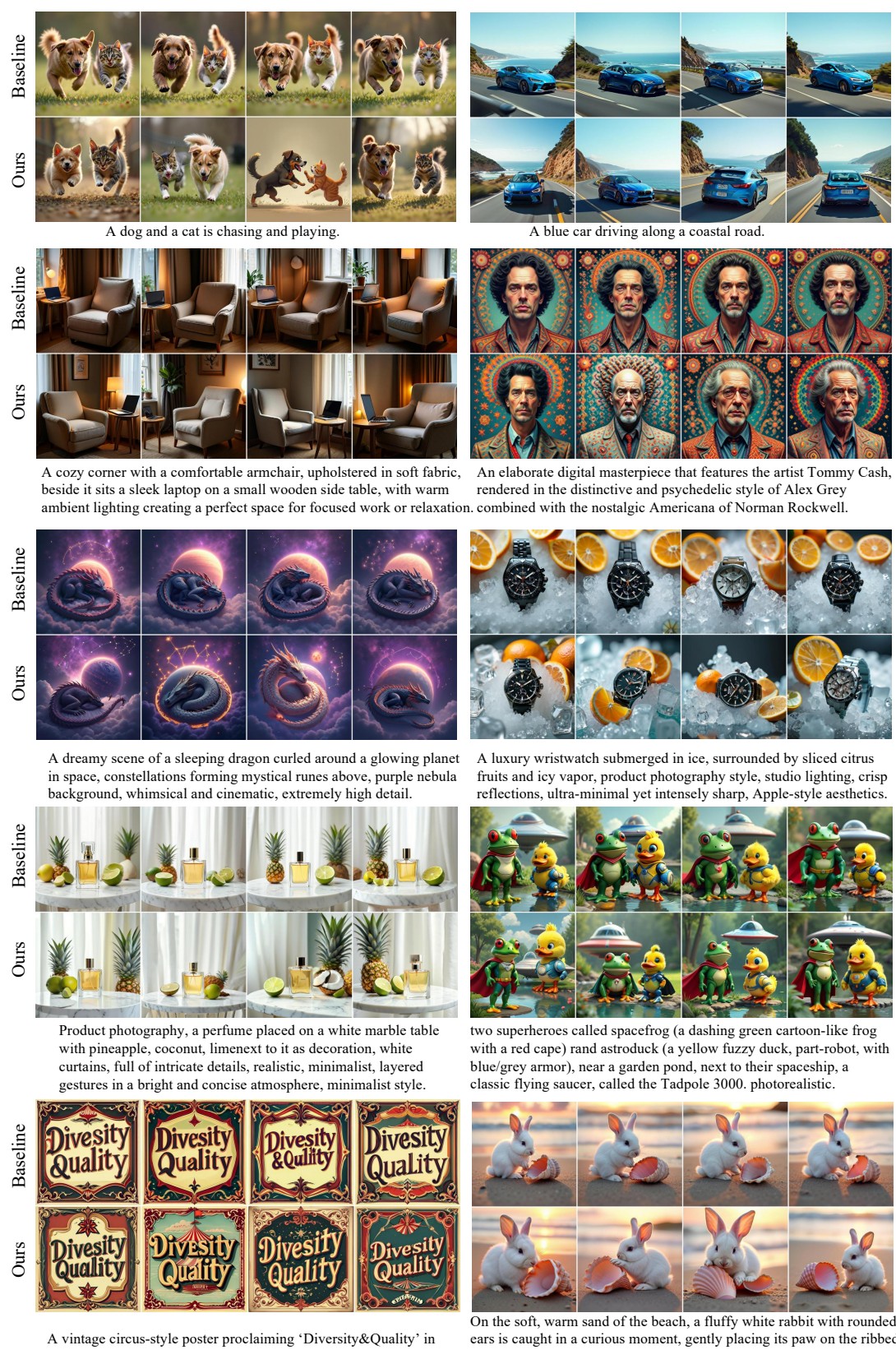

Figure 12: **Qualitative T2I comparison between original method and DiverseAR under the 2B model.** The first row shows baseline results; the second row shows DiverseAR results.

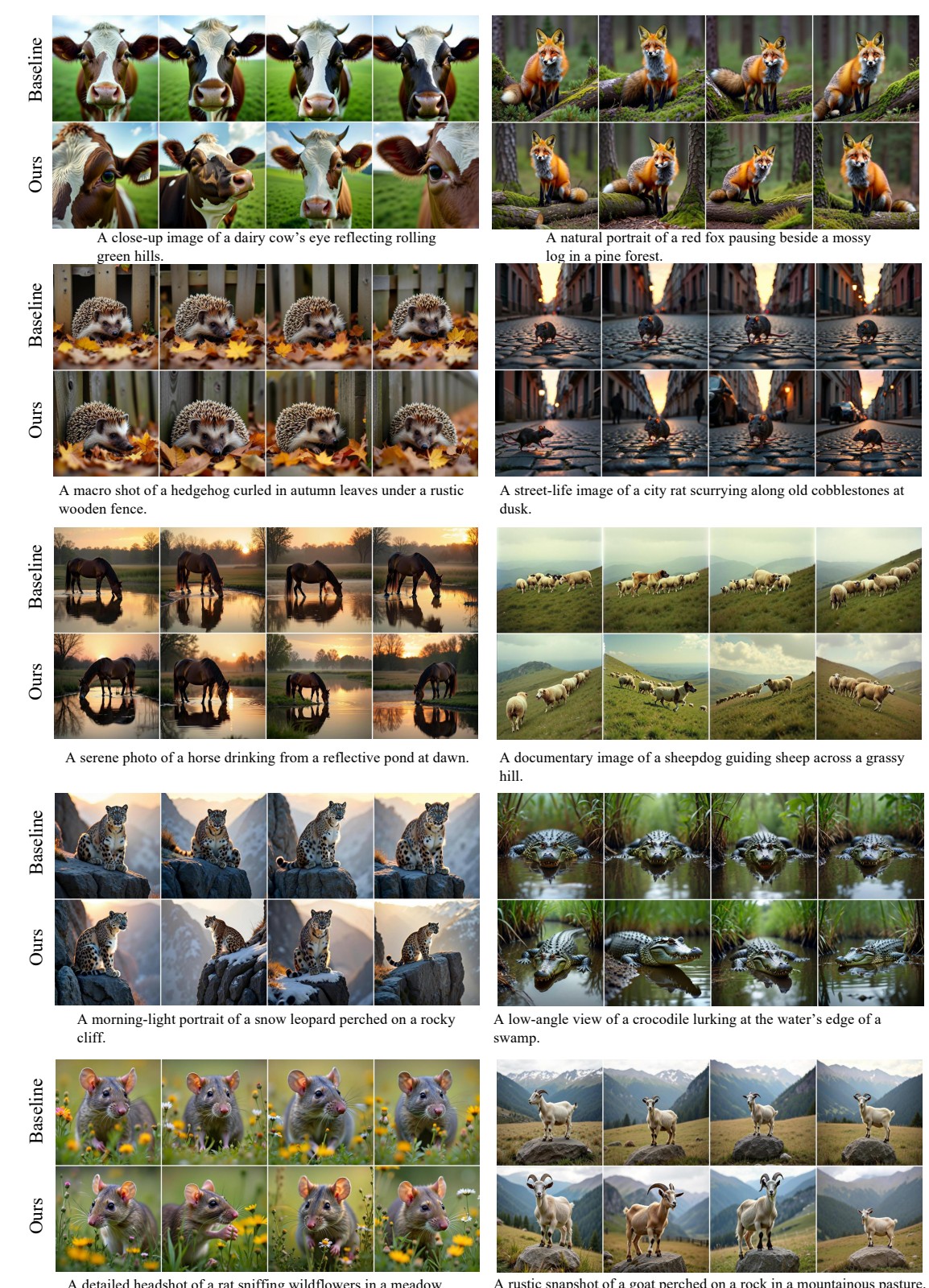

Figure 13: **More qualitative T2I comparison between original method and DiverseAR under the 2B model.** The first row shows baseline results; the second row shows DiverseAR results.

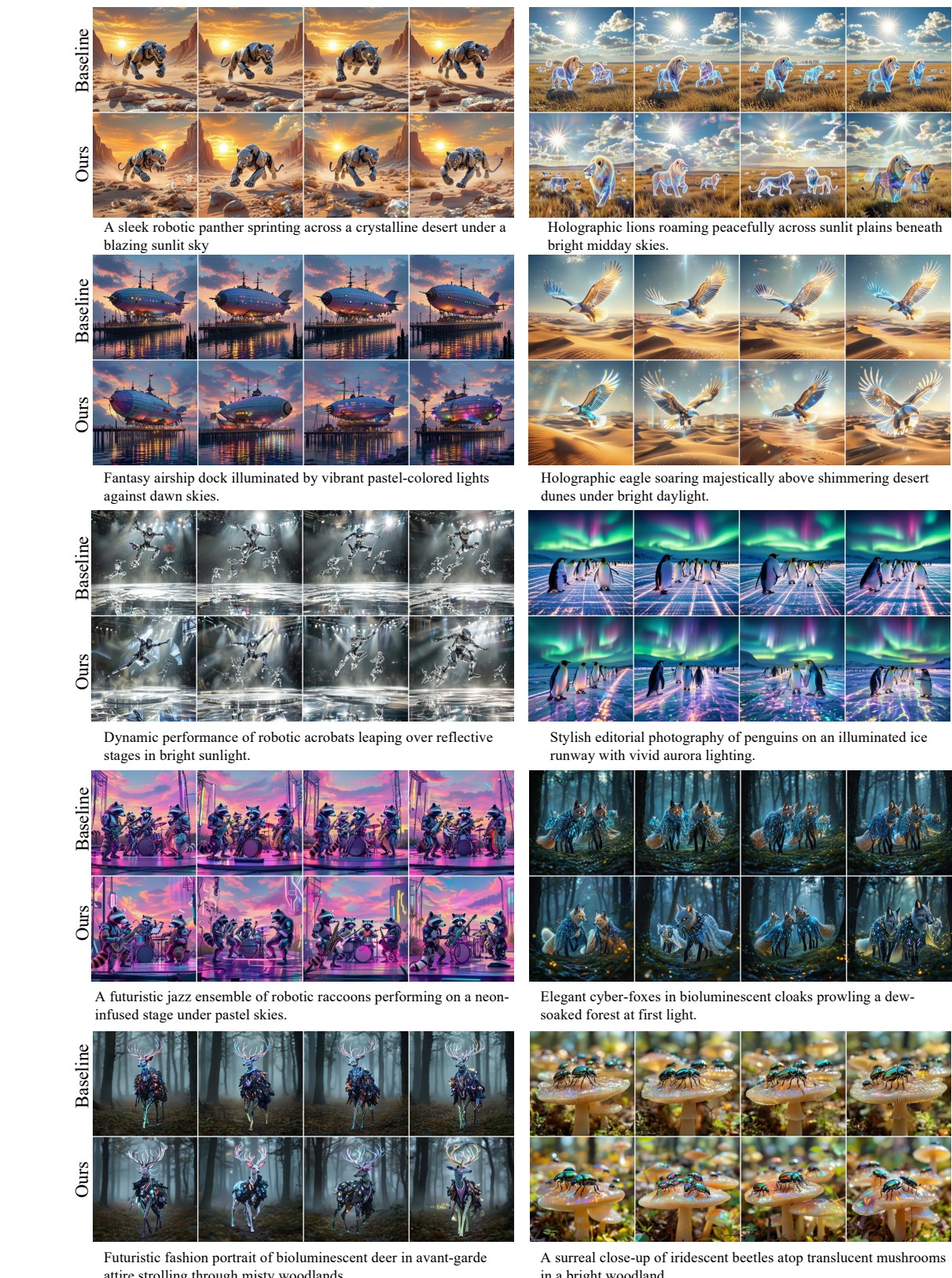

Figure 14: **Qualitative T2I comparison between original method and DiverseAR under the 8B model.** The first row shows baseline results; the second row shows DiverseAR results.

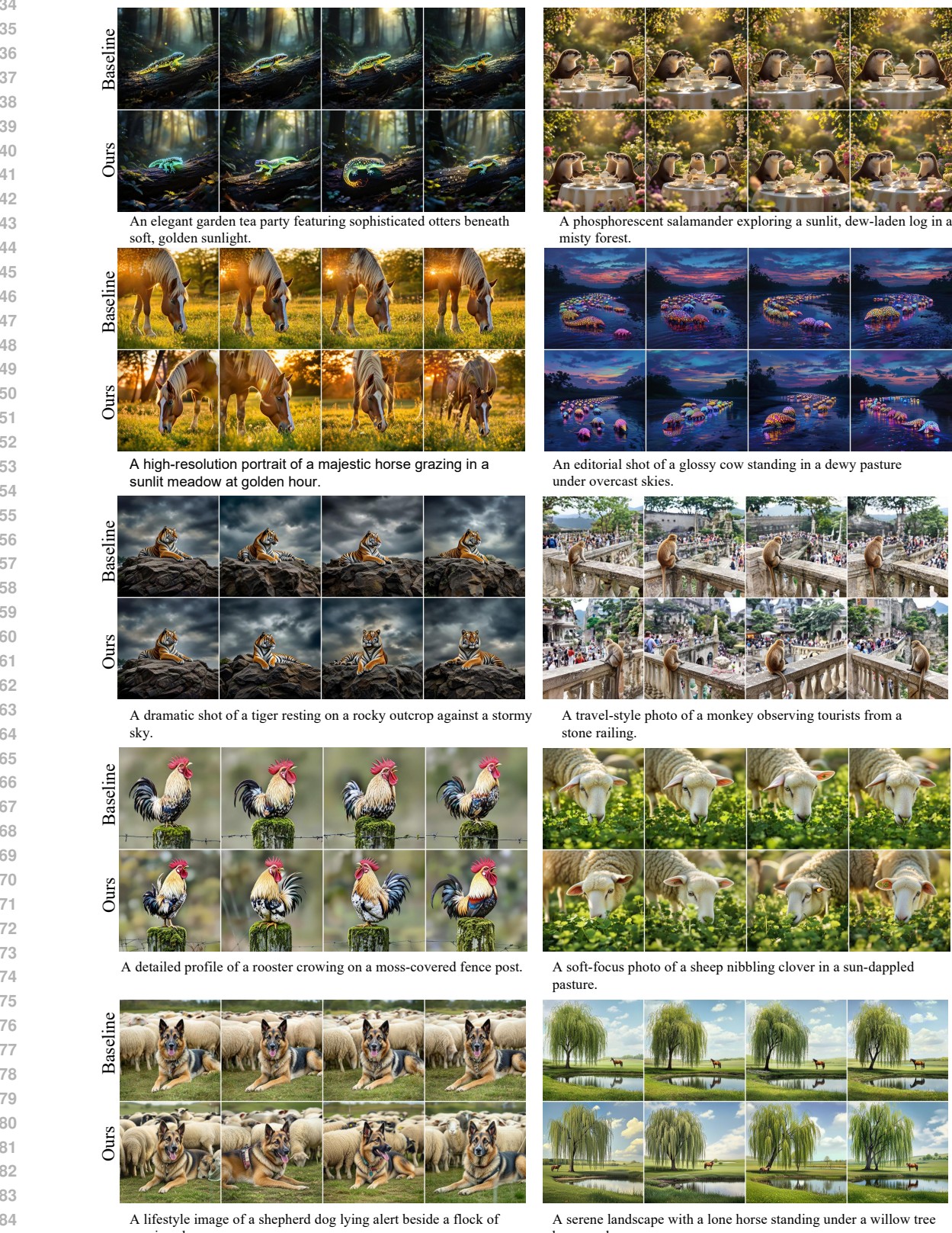

Figure 15: **More qualitative T2I comparison between original method and DiverseAR under the 8B model.** The first row shows baseline results; the second row shows DiverseAR results.

