# OpenReview forum: "DiverseAR: Boosting Diversity in Bitwise Autoregressive Image Generation"
_ICLR.cc/2026/Conference — ICLR 2026 Conference Withdrawn Submission_

### Official Review · Reviewer_AT9Z · 2025-10-25

**Soundness:** 3
**Presentation:** 3
**Contribution:** 3
**Rating:** 4
**Confidence:** 3

**Summary:**

This paper introduces DiverseAR, a framework designed to enhance image diversity without sacrificing visual quality in bitwise autoregressive image generation models. The first proposed technique is an adaptive temperature scaling mechanism across different generation scales, which improves diversity on early scales while maintaining visual quality on latter scales. As the first technique may lead to low-confidence regions and introduce semantic artifacts, an energy-based generation path search is then proposed to enhance visual quality. Experiments demonstrate that DiverseAR improves diversity while maintaining visual quality for bitwise autoregressive image generation models.

**Strengths:**

- The analysis in Section 3.2 provides insightful explanations for the phenomenon that bitwise autoregressive models exhibit limited diversity, revealing their tendency to become overconfident at early generation scales.
- The experiments and ablation studies are comprehensive and well-designed, clearly demonstrating that DiverseAR achieves superior diversity compared to baseline models.

**Weaknesses:**

- This paper uses "LPIPS" and "CLIP" as metrics to compare generation diversity. These notations are similar to the commonly used "LPIPS score" and "CLIP score" for visual quality comparison and may lead to confusion.
- For adaptive temperature scaling, the target of maximum bit probability is one possible choice, but other targets such as entropy could also be considered. Including an ablation study over different targets would provide a more comprehensive understanding of the scaling mechanism.
- The adaptive temperature scaling mechanism is designed to improve diversity, yet Table 2 shows that adaptive $\tau$ also achieves better visual quality than the baseline. It would be helpful to discuss or analyze why improving diversity can simultaneously enhance visual quality in this setting.
- In section 4.1 implementation details, the paper stated that "For the remaining ones, we use argmax sampling to select the highest-probability bit at each position." Why is it different from the baseline which uses "a fixed sampling temperature of 0.5"? Also, since the sampling methods are different, why is "Infinity-2B" in Table 1 achieves exactly the same metrics as Table 5 "number of selected secales = 0"?
- In Table 11, including cases with larger M and N would strengthen the ablation.

**Questions:**

Please refer to the weakness session.

---

### Official Review · Reviewer_bCNN · 2025-10-29

**Soundness:** 3
**Presentation:** 3
**Contribution:** 3
**Rating:** 6
**Confidence:** 4

**Summary:**

This paper investigates the prevalent problem of insufficient sample diversity in bitwise autoregressive (AR) image generation models such as Infinity. Through in-depth analysis, the authors attribute this limited diversity to two main reasons, and then conducted extensive experiments on the Infinity-2B and 8B models. The results show that DiverseAR significantly improves sample diversity while maintaining or even slightly improving the generated image quality and cue word-image alignment.

**Strengths:**

1.Strong and Comprehensive Empirical Validation
2. Clear Writing and Strong Logic

**Weaknesses:**

1. Energy-based pathfinding requires sampling $M$ paths and forward propagating $N$ steps, introducing additional computational overhead (approximately 1.12x according to Table 2). Although the authors refer to this as "minimal," it remains a factor to consider in large-scale inference deployments.

2. The core contribution of this paper lies primarily in the sampling function during the inference phase, without altering the model architecture or the training process itself. While addressing sampling diversity is an important issue, some reviewers might argue that optimizing the sampling algorithm alone offers a relatively limited contribution compared to works proposing new models or training paradigms.

**Questions:**

See Weakness below

---

### Official Review · Reviewer_fFFi · 2025-11-01

**Soundness:** 1
**Presentation:** 1
**Contribution:** 1
**Rating:** 0
**Confidence:** 1

**Summary:**

N/A.

**Strengths:**

N/A.

**Weaknesses:**

The authors have modified the official ICLR template by increasing the page margins (specifically, the sidebar spacing). This change effectively allows them to include more content than permitted under the standard format, which constitutes a clear formatting violation. Given that ICLR explicitly requires all submissions to strictly follow the provided style file and formatting guidelines, I believe this submission should be considered for desk rejection due to non-compliance with the conference format requirements.

**Questions:**

N/A.

---

### Official Review · Reviewer_NLik · 2025-11-13

**Soundness:** 1
**Presentation:** 1
**Contribution:** 1
**Rating:** 0
**Confidence:** 1

**Summary:**

N/A

**Strengths:**

N/A

**Weaknesses:**

It is clear that the submission does not comply with the ICLR 2026 formatting guidelines. The authors have intentionally and significantly reduced the top and side margins of the manuscript.

This modification allows them to fit substantially more content than permitted, effectively bypassing the page limit. This provides an unfair advantage over other authors who have adhered to the submission rules.

**Questions:**

N/A

---

### Note · Authors · 2025-11-13

I have read and agree with the venue's withdrawal policy on behalf of myself and my co-authors.